# Functional Characterization of a (*E*)-β-Ocimene Synthase Gene Contributing to the Defense against *Spodoptera litura*

**DOI:** 10.3390/ijms24087182

**Published:** 2023-04-13

**Authors:** Taotao Han, Yan Shao, Ruifang Gao, Jinshan Gao, Yu Jiang, Yue Yang, Yanan Wang, Siqi Yang, Xiang Gao, Li Wang, Yueqing Li

**Affiliations:** 1Key Laboratory of Molecular Epigenetics of MOE, Northeast Normal University, Changchun 130024, China; 2College of Plant Science, Jilin University, Changchun 130024, China; 3Key Laboratory of Soybean Molecular Design Breeding, Northeast Institute of Geography and Agroecology, Chinese Academy of Sciences, Changchun 130102, China

**Keywords:** anti-insects, (*E*)-β-ocimene, soybean, volatile terpenes

## Abstract

Soybean is a worldwide crop that offers valuable proteins, fatty acids, and phytonutrients to humans but is always damaged by insect pests or pathogens. Plants have captured sophisticated defense mechanisms in resisting the attack of insects and pathogens. How to protect soybean in an environment- or human-friendly way or how to develop plant-based pest control is a hotpot. Herbivore-induced plant volatiles that are released by multiple plant species have been assessed in multi-systems against various insects, of which (*E*)-β-ocimene has been reported to show anti-insect function in a variety of plants, including soybean. However, the responsible gene in soybean is unknown, and its mechanism of synthesis and anti-insect properties lacks comprehensive assessment. In this study, (*E*)-β-ocimene was confirmed to be induced by *Spodoptera litura* treatment. A plastidic localized monoterpene synthase gene, designated as *GmOCS*, was identified to be responsible for the biosynthesis of (*E*)-β-ocimene through genome-wide gene family screening and in vitro and in vivo assays. Results from transgenic soybean and tobacco confirmed that (*E*)-β-ocimene catalyzed by GmOCS had pivotal roles in repelling a *S. litura* attack. This study advances the understanding of (*E*)-β-ocimene synthesis and its function in crops, as well as provides a good candidate for further anti-insect soybean improvement.

## 1. Introduction

Soybean (*Glycine max*), a herbaceous annual plant in the family Fabaceae, is among the most important crops in the world, playing irreplaceable roles in providing humans and livestock with high-quality proteins and balanced fatty acids [1,2,3,4,5]. Phytonutrients such as saponins, vitamins, and flavonoids in soybean also contribute special health nutrition and pharmacological effects to humans [6,7,8,9]. The issue of how to guarantee or improve soybean yield and nutrition has attracted special attention from both botanists and breeders around the world. However, soybean diseases and pests have always been great threats to soybean production, which should never be ignored [10,11]. As the understanding of chemical pesticides deepens, studies on the environment or human-friendly pesticides or anti-pest crops have been a hotspot in recent years.

As sessile organisms, plants have to face complicated potential threats in their habitat and have evolved a robust immune system for their survival, such as trichomes defending against insects, specific metabolites conferring warning color and scent, or toxic properties to plants [12,13,14,15]. Terpenes are a group of plant-specialized metabolites with highly diverse structures and versatile applications, which have been reported to help plants counter various biotic and abiotic stresses [14,15,16,17,18,19,20,21,22,23,24,25,26]. Terpenes can further be classified as monoterpenes, sesquiterpenes, diterpenes, sesterterpenes, triterpenes, and tetrapterpenes/carotenoids, among which monoterpenes and sesquiterpenes are volatile terpenes playing indispensable roles in plant–insect communications such as to attract pollinators, seed dispersers or natural enemies of herbivores, or repel pathogens or herbivores directly [18,23,24,25]. Generally, the classification of terpenes is based on the numbers of fundamental repeating five-carbon isoprene units: isopentenyl diphosphate (IPP) and its double-bond isomer dimethylallyl diphosphate (DMAPP). IPP and DMAPP are synthesized through two compartmentalized pathways in plants: the plastidic methylerythritol 4-phosphate (MEP) pathway and the cytosolic mevalonic acid (MVA) pathway by sequential catalysis of a series of enzymes [15,27,28,29,30]. Subsequently, the sequential condensation of IPP units to one molecule of DMAPP by short-chain prenyltransferases (PTs) results in C10 geranyl diphosphate (GPP) or neryl diphosphate (NPP), C15 farnesyl diphosphate (FPP), C20 geranylgeranyl diphosphate (GGPP), etc., which are further catalyzed into versatile terpenes with rearrangements and cyclization reactions by a group of mechanistically related enzymes: terpene synthases (TPSs), whose copy number, subcellular localization, and catalytic property could largely determine the terpene diversity in a specific plant species.

Terpene synthases are encoded by a broad mid-sized *TPS* superfamily. Phylogenetically, land-plants-derived TPS proteins can be categorized into seven subfamilies, namely TPS-a, TPS-b, TPS-c, TPS-d, TPS-e/f, TPS-g, and TPS-h [29,31,32,33,34]. Specifically, TPS-d and TPS-h subfamilies contain TPSs exclusively found in gymnosperms and *Selaginella* spp., respectively. In comparison, TPSs responsible for volatile terpenes in angiosperms are confined into TPS-a, -b, and -g subfamilies. It has generally been observed that monoterpenes are mainly yielded by TPSs from the TPS-b clade, whereas sesquiterpenes are primarily produced by the TPS-a subfamily. The TPS-g subfamily includes both monoterpene synthases and sesquiterpene synthases, mainly represented by linalool synthase and nerolidol synthase, respectively. Moreover, TPSs can also be divided into two classes, Type I and Type II, based on the conserved motifs in either C or N termini, representing two diverse catalyzing mechanisms of terpene formation by TPSs [28,31]. It is widely accepted that the olfactory cues conferred by volatile terpenes, together with visualization, play important roles in the host identification of insects. It is also reported that many volatile terpenes resulting from TPSs of TPS-a, b, and g subfamilies can protect plants by either conferring antixenosis to kill insects or attracting their natural enemies [21,24,35,36]. However, the anti-GMO or some consumers may be prudent with transgenic soybeans expressing exogenous genes that confer terpene-based biopesticides. Herein, it is important to characterize soybean *TPS* genes functional in soybean protection from pests.

Research on volatile soybean terpenes has been conducted for decades. For example, Ricardo del Rosario et al. discovered the monoterpene α-pinene in wild soybeans in 1984 [37]. It is also found that (*E*)-β-ocimene and linalool might have anti-insect effects in insect-resistant soybean cultivars, but the underlying *GmTPS* genes remain to be uncovered [38]. In addition, (*E*,*E*)-α-farnesene is among the compounds with the greatest variation between damaged and undamaged soybean plants [11]. The advent of fast-developing omics sequencing tools has significantly accelerated modern studies such as gene characterization. It is reported that the soybean TPS family consists of more than 20 members in the soybean genome [39], of which *GmTPS5* and *GmTPS18* (*GmNES*) were reported to encode geraniol synthase and nerol synthase, respectively, to defend against cotton leaf worms (*Spodoptera litura*) in transgenic tobaccos [39,40]. In addition, *GmTPS21 (GmAFS)* was reported to encode α-farnesene synthase functioning in defense against nematodes and insects [24]. The *S. litura*, which is reported to infest 87 species of host plants with economic importance, is a major defoliator causing significant yield loss mainly by damaging the foliage [41]. In the present study, we re-mined the soybean *TPS* family and checked *TPS* genes responding to *S. litura*, methyl jasmonate (MeJA), and mechanical wounding treatments. Based on the genome-wide analysis of the *GmTPS* gene family, *GmTPS3* (later renamed as *GmOCS*) was subjected to further detailed analyses such as subcellular localization analysis, in vitro enzymatic assay, overexpression assays in soybean and tobacco, and dual-choice assay, demonstrating that *GmOCS* is an excellent candidate in further anti-insect soybean modification through controlling the biosynthesis of (*E*)-β-ocimene. Our findings provide valuable insights into understanding the mechanisms underlying defense responses of soybean against pests and also shed light on an environmentally friendly strategy for soybean pest control.

## 2. Results

### 2.1. Soybean TPS Gene Family Characterization

Initially, 43 sequences were mined by co-analysis of BLAST and HMM search from the *Glycine max* Wm82. a2. v1 genome database. After CD-Search and sequence length confirmation, 23 sequences were likely to encode full-length TPS proteins (Appendix A), which was also reported by Liu et al. in 2014 [39]. To better define the 23 *GmTPSs*, they were named following Liu et al. in 2014 [39], and a phylogenic tree was constructed together with several AtTPSs. Results showed that GmTPSs fell into different subfamilies, as narrated in Introduction, implying their functions might be divergent in terpene biosynthesis (Figure 1A). Gene structure is another important index defining the evolution or divergence of multigene family members. Consequently, an exon-intro diagram of soybean *GmTPSs* was also constructed to detect the structural diversity. The number of introns ranged from six to fourteen in *GmTPSs*. Intriguingly, all the *GmTPSs* from TPS-a, TPS-b, and TPS-g subfamilies had six introns except that *GmTPS11* and *GmTPS12* had seven introns. Comparatively, *GmTPSs* from more ancient TPS-c and TPS-e/f subfamilies showed many more introns (Figure 1A). Furthermore, a total of 10 conserved motifs were tried to be predicted in the GmTPSs by MEME suite. Results showed that GmTPSs from TPS-a, TPS-b, and TPS-g subfamilies shared similar motif patterns (Figure 1A). Together, the intron/extron pattern and conservative motif analysis implied the potential diverse functions of GmTPSs in different phylogenetic groups. Moreover, the chromosomal locations of the 23 *GmTPS* genes belonging to TPS-a, b, and g subfamilies were obtained from the soybean genome database. Results showed that the 23 *GmTPS* genes were distributed on nine chromosomes accounting for nearly half of all the 20 chromosomes (Figure 1B). Obviously, there are eight *GmTPSs* on chromosome 12, which had the highest density. Specifically, we mainly focused on the 18 GmTPSs belonging to TPS-a, TPS-b, and TPS-g subclades which were well-known enzymes responsible for volatile monoterpene and sesquiterpene biosynthesis in angiosperms. The transcripts of 18 *GmTPSs* in different soybean tissues or organs of Wm82 were checked with earlier published RNA-seq data [42]. Conspicuously, trace transcripts of most *GmTPSs* were detected in these samples. In contrast, *GmTPS12*, *GmTPS20*, and *GmTPS21* were highly expressed in leaves (Figure 1C). Herein, the results indicated that most *GmTPS* genes might be responsive to elicitors as stress-induced genes. Alternatively, some of the unexpressed *GmTPS* genes might become pseudogenization during evolution.

### 2.2. Volatile Terpenes and GmTPSs Induced by MeJA, Mechanical Wounding, and Spodoptera litura Treatments

To further explore the biological roles of *GmTPSs* in response to different stimuli, MeJA, mechanical wounding, and *S. litura* were employed to treat soybean seedlings, followed by volatile detection and *GmTPS* expression analysis. As a result, linalool was the most conspicuously elevated terpene detected from MeJA-treated soybeans (Figure 2A). Concurrently, *GmTPS3* (later renamed as *GmOCS* in present study), *GmTPS4*, *GmTPS8*, *GmTPS12*, *GmTPS20*, *GmTPS21*, and *GmTPS23* were among the genes with increased transcripts (Figure 2A). Furthermore, mechanical wounding was performed to simulate the bite of herbivores. The increase of a series of volatiles was detected in different contents, among which limonene, pinene, and ocimene were the conspicuous ones. Simultaneously, *GmTPS4*, *GmTPS8*, *GmTPS16*, and *GmTPS19* were substantially activated (Figure 2B). In comparison, linalool, α-farnesene, and (*E*)-β-ocimene were largely released from soybean seedlings treated with *S. litura.* Furthermore, the transcripts of *GmOCS* and *GmTPS21* continuously increased with a prolonged attack by *S. litura* (Figure 2C). To better illustrate the volatiles emitted by the plants, (*E*)-β-ocimene was subsequentially calculated based on its standard curve, and other terpenes were quantified as content relative to (*E*)-β-ocimene (Appendix A). All the aforementioned results indicated that different *GmTPSs* might be activated by different stimuli, resulting in the increase of distinct terpene volatiles.

### 2.3. Sequence Analysis of GmOCS

As *GmOCS* and *GmTPS21* were the genes with the most substantially activated expressions in *S. litura* and MeJA-treated soybean, their promoters were subjected to further analysis. Results showed that both *GmTPS* genes might be responsive to several abiotic environmental cues such as light, drought, and phytohormones, including abscisic acid and gibberellin, etc. (Figure 3A). To better understand their roles in soybean stress response, *GmOCS* was selected and cloned from soybean cultivar Wm82 by routine PCR as *GmTPS21* was characterized to encode a sesquiterpene synthase, which was not able to produce (*E*)-β-ocimene. Online predictions indicated that *GmOCS* encoded a protein with a molecular weight (MW) of 68.7 KDa and a theoretical PI at 6.25, and the protein had an instability index [44] of 43.83, which was predicted to be localized in chloroplast, implying that GmOCS might function as a monoterpene synthase and should be degraded soon as an instant and temporary response to external stimuli (Figure 3B). Detailed amino acid alignment revealed that GmOCS contained terpene-cyclization-related RR(X)_8_W motif, Mg^2+^ or Mn^2+^ cofactor-binding-related DDXX (D/E) and (N,D) DXX (S,T,G) XXXE (NSE/DTE) motifs (Appendix A). Moreover, monoterpene synthases were generally predicted to play roles in plastids, whereas sesquiterpene synthases were usually localized in the cytoplasm. To further check the potential subcellular localization of GmOCS, the GmOCS-GFP fusion protein was expressed in *Arabidopsis* (*Arabidopsis thaliana*) protoplasts. Consequently, it was primarily confined in chloroplasts, indicating its possible role in monoterpene biosynthesis (Figure 3C).

### 2.4. Functional Characterization of GmOCS

To better characterize *GmOCS*, it was firstly subcloned into pET32a to prepare recombinant proteins. Soluble GmOCS protein was successfully induced by IPTG (Appendix A). Afterward, enzymatic assays were performed with four kinds of reported substrates: GPP, NPP, (*E*,*E*)-FPP, and (*Z*,*Z*)-FPP, respectively. Compared to the prokaryotic proteins from bacteria expressing empty vector, GmOCS catalyzed GPP mainly into (*E*)-β-ocimene (94.27%), (*Z*)-β-ocimene (3.47%), and linalool (2.26%) (Figure 4A, Appendix A). However, no obvious products were observed when GmOCS was incubated with NPP, (*E*,*E*)-FPP, or (*Z*,*Z*)-FPP.

To further validate the functions of *GmOCS*, we constructed a transient expression vector harboring *GmOCS* driven by *35S* promoter and transformed it into *A. tumefaciens* GV3101 which was sequentially infiltrated into *N. benthamiana* leaves. As a result, *N. benthamiana* leaves transiently overexpressing *GmOCS* mainly released (*E*)-β-ocimene (Figure 4B), validating the result concluded from in vitro enzymatic assays. Herein, GmOCS could tentatively be attributed as a monoterpene synthase or (*E*)-β-ocimene synthase exclusively.

### 2.5. Overexpressing GmOCS in Transgenic Soybeans Led to Enhanced Resistance to Spodoptera litura

To better investigate the *GmOCS* role in anti-herbivore, it was stably transformed into soybean P3 by routine Agrobacterium infection. Five transgenic soybean lines were selected for further analysis. However, no obvious phenotypic changes were observed (Figure 5A). RT-qPCR (reverse-transcription quantitative PCR) analysis validated that *GmOCS* had different expression levels in these lines (Figure 5B). Whereafter, the leaves were harvested and further analyzed by SPME-GC/MS (solid-phase microextraction gas chromatography/mass spectrometry). When compared to the control, the five transgenic lines overexpressing *GmOCS* released more (*E*)-β-ocimene and α-farnesene (Figure 5C and Appendix A). To further investigate the effects of transgenic soybeans overexpressing *GmOCS* on herbivores, *S. litura* was employed in the following dual-choice feeding preference test. Interestingly, more gaps were observed from the control leaves when compared with transgenic leaves, indicating that *S. litura* was repelled by leaves overexpressing *GmOCS* (Figure 5D). To quantify the deterrence, the amount of leaves eaten by *S. litura* was assessed, indicating that *GmOCS* might be a good candidate gene for repelling soybean herbivores represented by *S. litura* (Figure 5E).

### 2.6. GmOCS Is a Good Candidate in Anti-Herbivore Plant Modification

To further investigate the applicability of *GmOCS* in the deterrence of herbivores in other plants, it was also overexpressed in *N. tabacum* (cv. K326). Consistently, transgenic tobaccos had no obvious phenotypic changes, but PCR results validated the existence of *GmOCS* (Figure 6A). Distinctly, (*E*)-β-ocimene was the primary volatile terpene detected from leaves of transgenic plants by SPME-GC/MS analysis (Figure 6B,C), which was in accordance with the results of earlier transient *N. benthamiana* assay (Figure 4). A dual-choice feeding-preference test was also performed to check the effects of *GmOCS* overexpression in tobacco. A similar conclusion could also be summarized that the increase of (*E*)-β-ocimene resulting from *GmOCS* overexpression conferred significant deterrence against *S. litura* (Figure 6D,E).

## 3. Discussion

Safeguarding plants against pests, microorganisms, and weeds is always the focus of the plant protection field. Plants produce a large variety of specialized metabolites that can function as active anti-stress molecules, and how to explore and employ these natural molecules has attracted much attention. It is widely documented that terpenes are among the most reported bioactive compounds with anti-insect, anti-microbial (phytoalexin), and allelopathic activities [17,46,47]. Especially, plant volatile terpenes, mainly composed of monoterpenes and sesquiterpenes, play pivotal roles in plant protection, perhaps primarily by repelling insect herbivores, attracting the natural enemies of the herbivores, and increasing the resistance of plants by signaling within or between plants [25,48,49,50,51]. Intriguingly, the selective modulation nature that terpenes seem to be more toxic to the enemy insects but with only little effects on beneficial insects, making terpenes an amazing candidate for pesticide development [52]. However, research on herbivore-induced plant volatiles has mainly focused on model plants instead of agricultural crops. Among these, the monoterpene (*E*)-β-ocimene is perhaps one of the most common volatiles whose release is induced by herbivory and is thought to protect plants by attracting natural enemies or repelling the insects [23,53,54,55]. Since the first (*E*)-β-ocimene synthase gene (*AtTPS03*) was cloned from *Arabidopsis*, more and more *TPS* genes responsible for (*E*)-β-ocimene synthesis have been characterized, and their anti-insect functions in some plants, such as tomato, Chinese cabbage, and lima bean were also deciphered [56,57,58].

In the present study, soybean *GmOCS* was proved to encode (*E*)-β-ocimene synthase by both in vitro enzymatic assay and transient *N. benthamiana* overexpression assay, in which β-ocimene, especially (*E*)-β-ocimene, was the most obvious product (Figure 4). Elicitor-induced assay and dual-choice feeding-preference assay indicated that *GmOCS* was substantially activated by a *S. litura* attack and, in turn, repelled the insect with its product (*E*)-β-ocimene (Figure 2 and Figure 5). Generally, insects may damage plant leaves during an attack. Simultaneously, JA increases significantly and is proposed to induce an increase in β-ocimene [50,59,60]. Hereto, MeJA and mechanical wounding are generally employed to simulate the herbivore attack in these assays. Notwithstanding, they cannot be replaced with each other as different volatiles and gene responses were observed in our results (Figure 2). This could be understandable as even the bouquets of volatiles were different when referring to different herbivore attacks. For example, *Helicoverpa zea*-damaged *N. tabacum* released more (*E*,*E*)-TMTT (4,8,12-trimethyl-1,3,7,11-tridecatetraene) and (*E*,*E*)-α-farnesene than *H. virescens*-attacked tobacco, which released more α-pinene instead [61]. Moreover, the feeding of *Pieris brassicae* and *Brevicoryne brassicae* had reverse-inducing effects on (*E*,*E*)-TMTT and α-barbatene release [62]. As a parasitic wasp of *Heliothis virescens*, *Cardiochiles nigriceps* captured the ability to distinguish between *H. virescens* and *H. zea*-induced plant volatiles to locate its host efficiently [61]. Owing to the complexity and diversity of different herbivore-induced plant volatiles in different plants, it is reasonable to deduce that the different components and content of volatiles released by plants may be efficient but different languages in plant–plant/insect communications. However, how the plants distinguish different elicitors is still a mystery.

As herbivore-induced plant volatiles, many terpene compounds can be induced in different plants by an attack from different insects; they may be common and exhibit broad-spectrum insect resistance, such as β-ocimene which has already been reported to exhibit anti-insect functions in many plant species aforementioned [17,21,35,57,58,61,63,64]. There are also several conjectures about the underlying anti-insect mechanism of these aforementioned terpenes, such as direct defense against insects, attracting insect predators, and signaling within or between plants [48,49,50]. However, the mechanisms of different terpenes may be distinct and also may not function alone but be consecutive and interconnected with each other. For example, monoterpene-derived cyclopentanoid (iridoid) is incredibly bitter and may directly cause the denaturation of amino acids and nucleic acids, stifle insect growth and development by inhibiting prostaglandins and leukotrienes synthesis [65]. In comparison, the anti-insect function of some volatiles might be achieved through the upstream signal molecule JA [66]. It is also proven that β-ocimene released from tea (*Camellia sinensis*) leaves does not affect tea geometrid (*Ectropis grisescens* Warren) directly but may change the phytohormone synthesis by fumigating with tea leaves [67]. Although we have demonstrated that β-ocimene synthesized by GmOCS has an anti-insect function in both soybean and tobacco (Figure 5 and Figure 6), its accurate anti-insect mechanism remains unclear. Moreover, results from in vitro and in vivo functional assays indicated that GmOCS could not synthesize α-farnesene (Figure 4), which was reported to be another kind of anti-insect volatile terpene [24]. However, α-farnesene in transgenic soybean was substantially detected (Figure 5). Considering α-farnesene was also substantially induced in *S. litura*-attacked soybeans (Figure 2), we tentatively thought that the increment of β-ocimene might lure α-farnesene synthesis through an uncovered routing, by which the soybeans may acquire a quick and effective anti-insect capacity.

Based on the present study, we tentatively summarized a model interpreting terpene release of soybean responses to MeJA, mechanical wounding, and *S. litura* treatments (Figure 7). However, there are still a series of issues remaining to be resolved, such as the accurate, responsive mechanisms underlying different treatments, the potential genes responsible for other volatiles, the effects of volatile terpenes on other insects and pathogens, and so on. To summarize, our study here lays the foundation for further understanding the anti-insect properties of soybean-volatile terpenes and paves the way to future anti-insect crop improvement.

## 4. Materials and Methods

### 4.1. Plant Materials and Growth Conditions

*Soybean* (Williams 82 and P3) plants were cultivated under greenhouse conditions with a photoperiod of 12 h light and 12 h dark at 22 °C. *A. thaliana* ecotype Columbia (*Col-0*), *N. benthamiana*, and *N. tabacum* (cv. k326) were grown in a greenhouse at 22 °C with a photoperiod of 16 h/8 h (light/dark).

### 4.2. Soybean Treatment

Four-week-old soybean seedings were treated with MeJA, mechanical wounding, and *S. litura*, respectively, following other studies with some modifications [24,39]. In detail, the soybean plants were sprayed either with 20 mL of 5 mM MeJA in ethanol or by ethanol. As for the mechanical wounding assay, soybean leaves were injured by a puncher with a diameter of 5 mm. Around 5~7 holes were made in each leaf. As for *S. litura* treatment, a total of 20 third instar larvae of *S. litura* were bought from Baiyun Industry Co., Ltd. (Jiyuan, Henan, China) and starved for 2 h before being further used to infect one soybean plant. Afterward, all the seedlings were covered by transparent covers, and volatiles were detected at 6, 12, or 24 h after treatment. Simultaneously, soybean leaves at each time point were harvested, immediately frozen into liquid nitrogen, and stored at −80 °C.

### 4.3. Volatile Terpene Detection

Headspace SPME-GC/MS was employed to examine the volatile terpenes following our earlier studies [68,69,70]. Briefly, the 100 mm DVB/CAR/PDMS fused silica fibers (Sigma-Aldrich, St. Louis, MI, USA) were used to absorb the volatiles at 22 °C for 2 h. The volatiles captured by the fibers were subsequently thermally desorbed and analyzed by an Agilent 5975-6890N GC-MS apparatus (Agilent Technologies, Santa Clara, CA, USA) equipped with a HP-1MS fused-silica capillary column. Volatile terpenes were identified by comparing the mass spectra and retention times with those of standard samples or compounds deposited in the NIST 2008 mass spectra library. The relative content of volatile terpenes was quantitatively analyzed according to the standard curve of (*E*)-β-ocimene.

### 4.4. RNA Extraction, cDNA Synthesis, and RT-qPCR Analysis

Total RNA was extracted from different samples following the manufacturer’s standard protocol of the OminiPlant RNA Kit containing DNase I (CWBIO, Beijing, China). The quality and concentration of RNA samples were assessed by Agarose gel and NanoDrop 2000 (ThermoFisher Scientific, Waltham, MA, USA). Afterward, 500 ng of total RNA was mixed with Oligo (dT)15 primer and random hexamer primer and reversed by M-MLV Reverse Transcriptase (Promega, Madison, WI, USA) following the instruction. Gene expression levels were evaluated with specific primers listed in Appendix A by RT-qPCR following our earlier studies [7]. The relative transcripts were normalized by *GmACTIN* (Glyma.19G147900.1) and *β-tubulin* (Glyma.03G124400.1) and calculated by 2^-ΔΔCT^ [45]. All the data were obtained from at least three biological replicates. The data were further subjected to TBtools and normalized by log_2_(FPKM + 1) for heatmap construction [45].

### 4.5. Soybean TPS Cloning and Sequences Analysis

Specific primers were designed to clone the soybean *GmOCS*, and the PCR result was cloned into the *pESI-T* vector provided in Hieff Clone^®^ Zero TOPO-TA Cloning Kit (Yeasen Biotech Co., Ltd., Shanghai, China) for sequencing confirmation. Soybean and *Arabidopsis* TPSs were subjected to the online Clustal Omega service (http://www.ebi.ac.uk/Tools/msa/clustalo/) (accessed on 6 January 2023) with defaulted parameters for sequence alignment. The alignment was subsequently processed by MEGA Ⅹ [71] to construct a neighbor-joining tree with 1000 bootstraps. Sequences with the following GenBank accession numbers were used in either phylogeny or alignment analysis: AtTPS10 (ACF41947), AtTPS14 (NP001185286), AtTPS21 (NP001190374), AtGA1 (Q38802.1), AtGA2 (Q9SAK2.1), AtTPS3 (NP_173143.4), PtTPS2 (AEI52902.1), MdAFS (AAO22848.2), CtGES (CAD29734.2), PlOS (ABY65110), LjEβOS (AAT86042), and MtTPS4 (Q5UB07). *GmTPSs* exon/intron structures were analyzed based on the genome annotation file downloaded from Phytozyme v13 (https://phytozome-next.jgi.doe.gov/) (accessed on 8 January 2023). The conserved motifs were predicted by the MEME Suite 5.5.1 (https://meme-suite.org/meme/tools/meme) (accessed on 6 February 2023) with default parameters, except for a maximum of 10 motifs selected. Both gene structures and conserved motifs were visualized with Gene Structure View (Advanced) plugin in TBtools [43]. The chromosomal locations of the *GmTPS* genes were presented with Gene Location Visualize from GTF/GFF plugin integrated with TBtools [43]. The cis-acting regulatory element in promoter sequences was analyzed by PlantCARE (https://bioinformatics.psb.ugent.be/webtools/plantcare/html/) (accessed on 16 February 2023) and visualized by Simple BioSequence Viewer in TBtools [43]. GmOCS sequence properties were analyzed with the ExPASy Protparam tool (https://web.expasy.org/protparam/) (accessed on 18 February 2023) and Plant-mPLoc server (http://www.csbio.sjtu.edu.cn/bioinf/plant-multi/) (accessed on 26 February 2023) with default parameters.

### 4.6. Vector Construction

Vectors used in the present study were constructed with the Minerva Super Fusion Cloning Kit (US EverbrightR Inc., Suzhou, China). Briefly, homologous arms were designed adjoining specific primers according to the vectors used. For vectors used in the prokaryotic expression of recombinant proteins, the coding sequence (CDS) of *GmOCS* was subcloned into *Bam*HI and *Eco*RI digested pET32a vector. For the vector used in subcellular localization, the CDS was subcloned into the earlier-used pUC19-HA-*GmMYBA2*-GFP by *Nde*Ⅰ and *Cla*Ⅰ [7]. For the vector used in the transient infiltration of *N. benthamiana*, the CDS was seamlessly cloned into pEAQ-HT by *Nru*Ⅰ and *Sac*Ⅰ [72]. For the vector used in the soybean and tobacco transformation, pTF101.1 was employed to contain the CDS driven by the 35S promoter and terminated by the NOS terminator. All the other vectors used in this study could be found in our earlier studies [7].

### 4.7. Transient Protoplast Assay

Plasmids were purified from *E. coli* JM109 using GoldHi EndoFree Plasmid Maxi Kit (CWBIO, Beijing, China) and enriched to ~4000 ng µL^−1^ with 5 M NaCl solution and isopropanol before transfection [73,74]. Protoplasts were isolated from rosette leaves of four-week-old *Arabidopsis* and further transfected by PEG3350. After 22 h incubation, the protoplasts were visualized by fluorescence microscopy (Olympus, Tokyo, Japan). Detailed information could be found in our earlier studies [75].

### 4.8. In Vitro Enzymatic Assay

Detailed methods for the prokaryotic expression of recombinant proteins and enzymatic assay have been introduced in earlier studies [76,77]. Briefly, the pET32a vector containing *GmOCS* was transformed into *E. coli* BL21 (DE3) and inducted by IPTG (isopropyl-β-D-thiogalactopyranoside). The cells were pelleted and sonicated to release recombinant protein, which was further purified by Ni-TED Sefinose™ column (Sangon Biotech, Shanghai, China) and analyzed by SDS-PAGE (Shanghai Epizyme Biomedical Technology Co., Ltd., Shanghai, China). Subsequently, 40~50 μg of recombinant protein was mixed with 15 mM MgCl_2_, 5 mM dithiothreitol, and 2 mM GPP, NPP, (*E*,*E*)-FPP or (*Z*,*Z*)-FPP (all purchased from Sigma Aldrich, St. Louis, MI, USA) in a total volume of 100 μL 25 mM HEPES buffer. The aforementioned DVB/CAR/PDMS fiber was employed to absorb the volatiles emitted from the reaction mixture before GC-MS analysis.

### 4.9. Plant Transformation

The constructed serial vectors based on pTF101.1 and pEAQ-HT backbone aforementioned were transformed into *Agrobacterium tumefaciens* GV3101. The transient transfection of *N. benthamiana* leaves has been described in detail in our earlier studies [70]. Four days after inoculation, the infiltrated leaves were cut into pieces and collected into an odor-free bottle to detect the volatiles. Agrobacterium-mediated *N. tabacum* stable transformation was conducted according to the procedures described in Sparkes et al., 2006 [78]. The regenerated plants were transferred to the greenhouse for the following gene expression, volatile detection, and dual-choice feeding-preference assays. As for soybean transformation, the seeds of soybean cultivar P3 were cultured on B5 media (Coolaber Biotech, Beijing, China) overnight. Then, the two cotyledons were separated and infected by the *GV3101* strain containing pTF101.1-*GmOCS* or control vector. After co-cultivation, shoot induction and selection, shoot elongation, plant rooting, plant hardening, and screening, the transgenic plants were subjected to further gene expression, volatile detection, and dual-choice feeding-preference experiments. Detailed information has been reported in earlier studies [18,39].

### 4.10. Dual-Choice Feeding-Preference Assay

The dual-choice feeding-preference tests were performed according to earlier reports with some modifications [39]. In short, mature full green leaves of similar size from transgenic soybean, tobacco, and control plants were placed on moist filter paper in a big Petri dish (100 mm × 100 mm). Twenty insects of third instar larvae stage of *S. litura* were released in the middle of the container. The remaining leaflet amount was measured after 12 h treatment to indicate the feeding preference.

## 5. Statistical Analysis

SPSS (version 23.0; IBM, Armonk, NY, USA) was used for statistical analyses, and data were expressed as mean ± standard deviation from at least three biological replicates, and a student’s *t*-test was used to analyze the significance of differences, which was considered statistically significant at *p* < 0.05 (** *p* < 0.01; * *p* < 0.05).

## Figures and Tables

**Figure 1 ijms-24-07182-f001:**
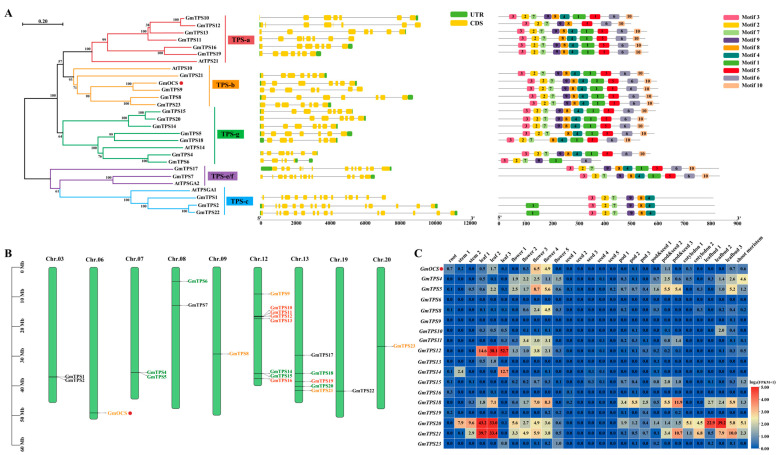
**Phylogeny, gene structure, conservative motif, gene location, and expression patterns of *GmTPS* genes**. (**A**) Phylogeny (left panel), gene structure (middle panel), and conservative motif (right panel) analysis of GmTPS sequences. The TPS sequences were processed by Clustal Omega and subjected to MEGA-X to construct the neighbor-joining tree. *GmTPS3* was renamed *GmOCS* later. Bootstrap values were 1000 replicates. Gene structures and conservative motifs were drawn using TBtools [43]. (**B**) Chromosomal distributions of *GmTPSs*. Different subcladed *GmTPSs* were characterized in different colors. (**C**) Transcripts of *GmTPSs* in different tissues or organs of soybean. The data were calculated with earlier published RNA-seq data [42]. The FPKM values were normalized by log_2_ and illustrated by TBtools with HeatMap Illustrator [43].

**Figure 2 ijms-24-07182-f002:**
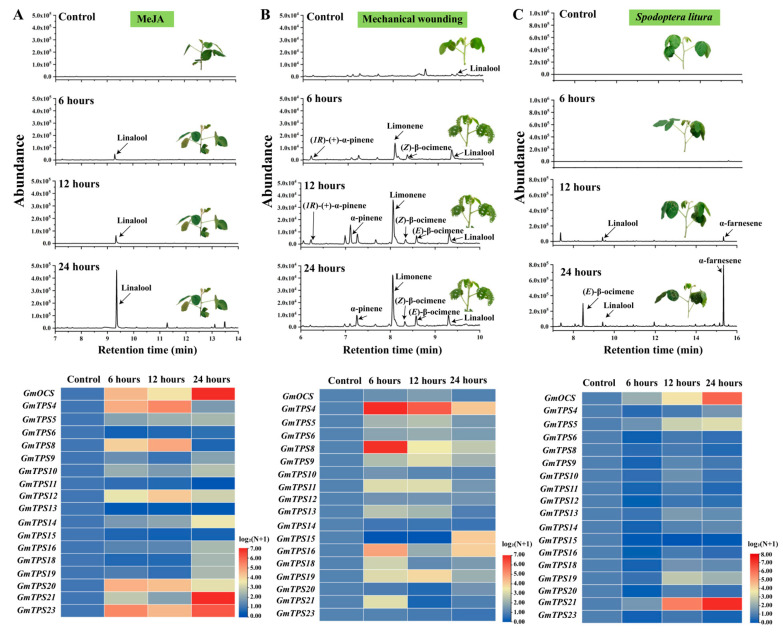
**Volatile terpenes and *GmTPS* expressions response to MeJA, mechanical wounding, or *S. litura* treatment**. (**A**) Volatile terpenes and expressions of *GmTPS* genes detected in soybean leaves treated with MeJA. (**B**) Volatile compounds and expression of *GmTPS* genes detected in soybean leaves under mechanical wounding condition. (**C**) Volatile compounds and expression of *GmTPS* genes induced by *S. litura*. Healthy and intact leaves without damage were used as control. Each experiment was performed in three biological replicates, and only representative data were provided. The transcripts were normalized by *GmACTIN* and *β-tubulin* and compared with the expression level of a specific gene in control. All the data were calculated as log_2_(N + 1). Red and blue boxes indicated high and low expression levels, respectively.

**Figure 3 ijms-24-07182-f003:**
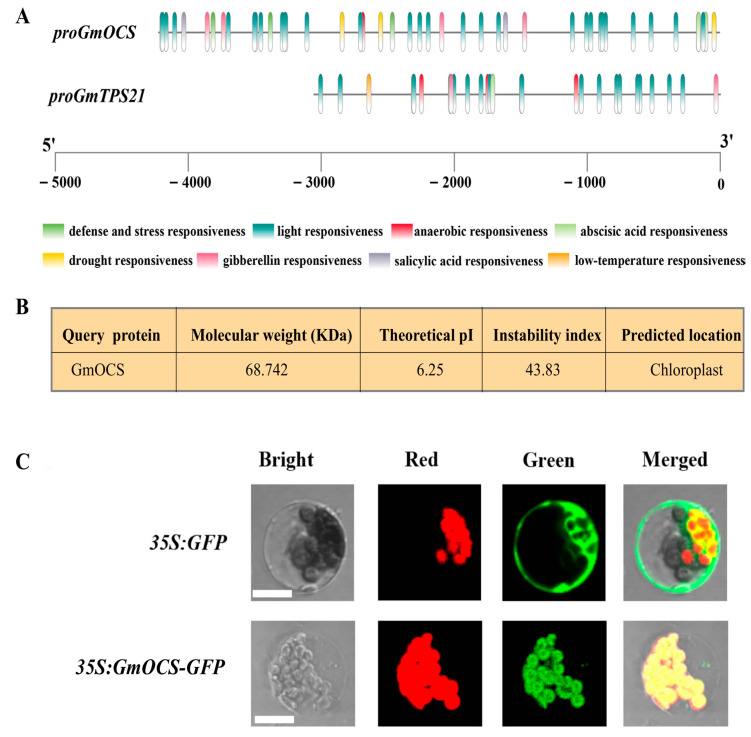
**Sequence and subcellular analysis of GmOCS.** (**A**) The *cis*-acting regulatory elements in promoter sequences of *GmOCS* and *GmTPS21*. The 4254 bp and 3052 bp upstream sequences of the initiation codon ATG were selected as promoter sequences of *GmOCS* and *GmTPS21*, respectively, and were analyzed by PlantCARE followed by being visualized with Simple BioSequence Viewer in TBtools [45]. (**B**) Basic sequence information of GmOCS. GmOCS sequence properties were analyzed with the ExPASy Protparam tool and Plant-mPLoc server with default parameters. (**C**) Subcellular localization of GmOCS in *Arabidopsis* protoplasts. Green, GFP fluorescence; Red, chlorophyll fluorescence; Bright Light, brightfield image; Merged, merged green, red, and bright light images. Scale bar = 25 μm.

**Figure 4 ijms-24-07182-f004:**
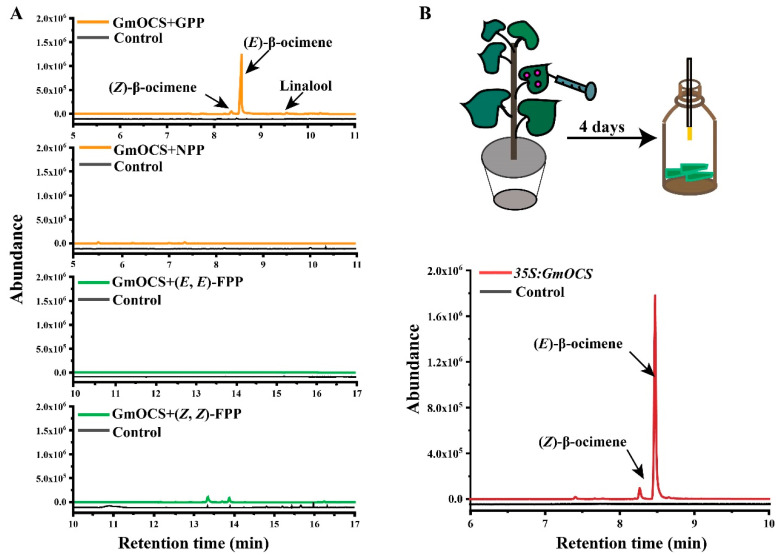
**In vitro and in vivo characterization of GmOCS.** (**A**) In vitro enzymatic analysis of GmOCS using four acyclic prenyl diphosphate substrates. X-axis represented retention time, and the y-axis represented product abundance. The contents of enzymatic products were detailed in Appendix A. (**B**) Volatile terpenes released from *N. benthamiana* leaves transiently overexpressing *GmOCS*. Agrobacterium containing *GmOCS* in pEAQ-HT backbone was infiltrated into *N. benthamiana* leaves. The leaves were sampled and analyzed by SPME-GC/MS analysis. *N. benthamiana* leaves infiltrated by *Agrobacterium* containing empty pEAQ-HT were used as control.

**Figure 5 ijms-24-07182-f005:**
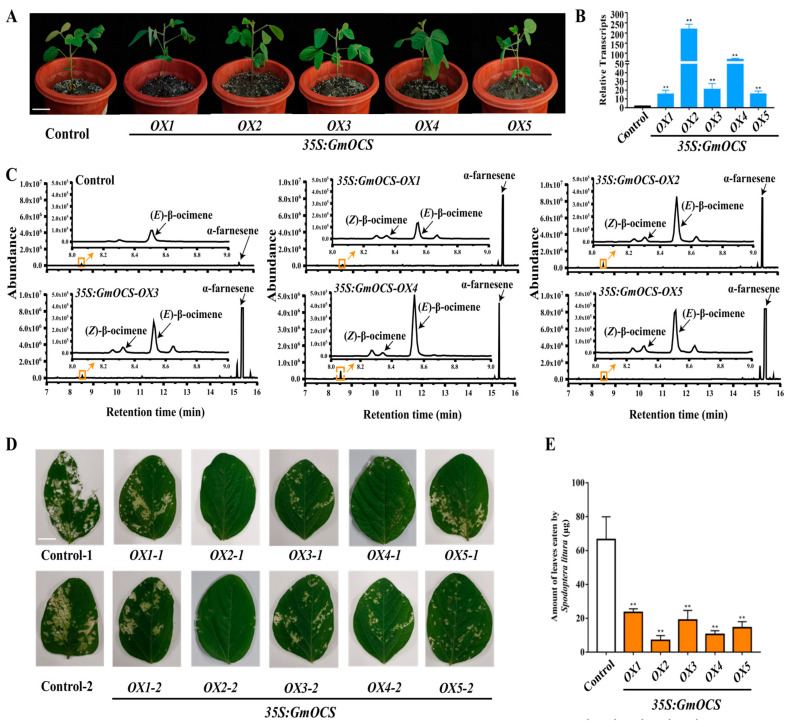
**The overexpression of *GmOCS* in soybean plants led to enhanced resistance against *S. litura***. (**A**) Control and transgenic soybean plants overexpressing *GmOCS*. (**B**) The relative expression levels of *GmOCS* in transgenic soybean leaves compared with the control. (**C**) Volatile terpenes detected in the control and transgenic soybean leaves overexpressing *GmOCS*. (**D**) *S. litura* attacked wild leaves and transgenic leaves, overexpressing *GmOCS* in a dual-choice feeding-preference experiment. (**E**) Leaves of wild and *GmOCS* overexpressed lines eaten by *S. litura* in dual-choice feeding-preference experiment. Data represented the mean ± SD of three biological replicates, and the student’s *t*-test was used to analyze the significance of differences (** *p* < 0.01) in (**B**,**E**).

**Figure 6 ijms-24-07182-f006:**
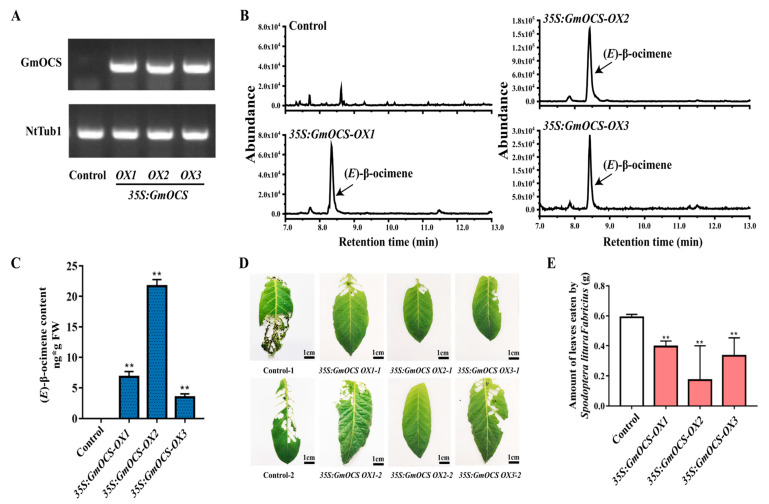
**The overexpression of *GmOCS* in *N. tabacum* led to enhanced resistance against *S. litura*.** (**A**) Expression analysis of the *GmOCS* gene by RT-PCR in the wild type and transgenic tobaccos. (**B**) The terpene release pattern detected from the control and transgenic tobacco leaves overexpressing *GmOCS*. (**C**) Relative contents of volatile terpenes detected in the control and transgenic soybean leaves overexpressing *GmOCS*. (**D**) *S. litura* attacked wild leaves and transgenic leaves, overexpressing *GmOCS* in a dual-choice feeding-preference experiment. (**E**) Leaves of wild and *GmOCS* overexpressed lines eaten by *S. litura* in a dual-choice feeding-preference experiment. Data represented the mean ± SD of three biological replicates, and the student’s *t*-test was used to analyze the significance of differences (** *p* < 0.01) in (**B**,**E**).

**Figure 7 ijms-24-07182-f007:**
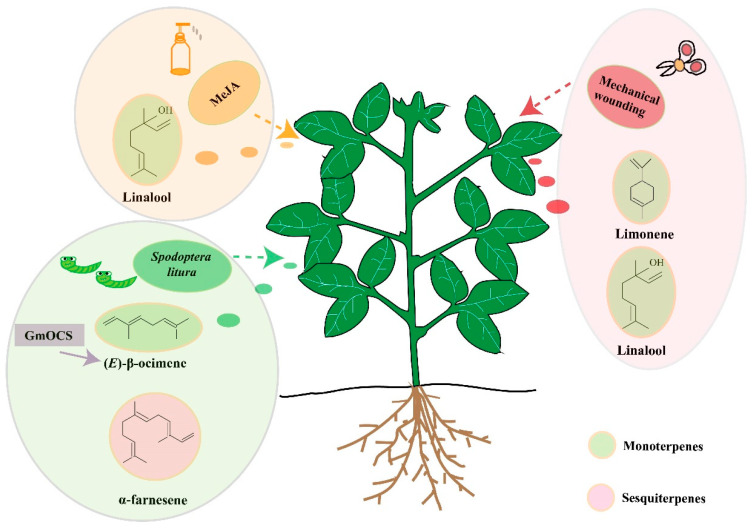
**Proposed model of soybean terpene releases responded to MeJA, mechanical wounding, and *Spodoptera litura* treatments**. Chemical molecules highlighted by green or pink background represented monoterpenes and sesquiterpenes, respectively.

## Data Availability

Data is contained within the article or Appendix A. For other information please contact the corresponding author.

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
