# Peer review of "Functional Characterization of a (E)-β-Ocimene Synthase Gene Contributing to the Defense against Spodoptera litura"

_ijms, 2023, doi:10.3390/ijms24087182_

Round 1

Reviewer 1 Report

This is a comprehensive analysis of an ocimene synthase gene from soybean in response to insect attack. In this study, the authors analyzed the terpene synthase gene family, screened candidate genes using different treatments and volatile analysis, performed functional characterization, and also tested the function of the gene in transgenic plants for volatile emission and insect resistance. The overall quality of this manuscript is high, and it is a pleasure to review this work. This reviewer has only a few minor comments as following:

1. Lines 50-51. Please decide to use either diphosphate or pyrophosphate throughout the manuscript, but not both.

2. Line 120. It should be "many more" I think.

3. Figures are of low resolution. I wish this is only because the conversion from word to PDF files, and the final version will have better ones.

4. Figure 1. Please mark or indicate GmOCS clearly (such as underlining or in bold).

5. Lines 182-183. Usually we don't list light and drought as elicitors. There are abiotic environmental cues.

6. Lines 18-189, please provide a reference for instability index, which is not frequently reported in related studies.

7. Figure 2B. I cannot see which sequences are used for alignment, because of the figure resolution. However, I think an alignment of a few more functionally characterized monoterpene synthases (from different plant species) can be a good supplemental figure. An annotation of the GmOCS protein sequence only can be used here as Figure 2B. I am not sure whether the authors predicted the putative chloroplast transit peptide sequence, which is normally performed for the "prokaryotic expression of recombinant proteins (line 217, not "prokaryotic protein)". If so, please also make the residue where the transit peptide starts, and also address this issue in Methods.

8. Line 227. I think the authors wanted to say "subsequently".

Reviewer 2 Report

An interestingly written article on protecting soybeans from pests. Plants are known to have developed sophisticated defense mechanisms to resist attack by insects and pathogens. It is important to find ways to protect soybeans in an environmentally or human-friendly way, or how to develop a plant-based method of pest control. More and more attention is being paid to volatile substances released by various plant species. Of these, (E)-β-ocimene was found to have anti-insect activity in various plants, including soybean. The authors confirmed that (E)-β-ocimene is induced by treatment of Spodoptera litura. A plastid localized monoterpene synthase gene, designated GmOCS, responsible for the biosynthesis of (E)-β-ocimene was identified by genome-wide gene family screening. Results from transgenic soybean and tobacco confirmed that GmOCS-catalyzed (E)-β-ocimene plays a key role in repelling S. litura attack. These results can be the basis for further research. The experiments were performed in a clear and accurate manner. Discussions were conducted exhaustively, including quite an extensive number of cited articles. I miss the material and methodological description of the statistical analysis in the chapter (it was mentioned only in Figure 6, about the t-test and statistically significant differences).

Figure 1. Poorly visible graphs. Maybe enlarge the drawings and put them in supplementary materials?
